

# Nomograms for predicting overall survival and cancer-specific survival in young patients with pancreatic cancer in the US based on the SEER database

Min Shi[*], Biao Zhou[*] and Shu-Ping Yang

Department of Gastroenterology, Liyang Branch of Jiangsu Province Hospital, Liyang, China
[*] These authors contributed equally to this work.

## ABSTRACT

**Background.** The incidence of young patients with pancreatic cancer (PC) is on the rise, and there is a lack of models that could effectively predict their prognosis. The purpose of this study was to construct nomograms for predicting the overall survival (OS) and cancer-specific survival (CSS) of young patients with PC.

**Methods.** PC patients younger than 50 years old from 2004 to 2015 in the Surveillance, Epidemiology, and End Results (SEER) database were selected and randomly divided into training set and validation set. Univariable and forward stepwise multivariable Cox analysis was used to determine the independent factors affecting OS. The Fine and Gray competing risk regression model was used to determine the independent factors affecting CSS. We used significant variables in the training set to construct nomograms predicting prognosis. The discrimination and calibration power of models were evaluated by concordance index (C-index), calibration curve and 10-flod cross-validation.

**Results.** A total of 4,146 patients were selected. Multivariable Cox analysis showed that gender, race, grade, pathological types, AJCC stage and surgery were independent factors affecting OS. The C-index of the nomogram predicting OS in training and validation was 0.733 (average = 0.731, 95% CI [0.724–0.738]) and 0.742 (95% CI [0.725–0.759]), respectively. Competing risk analysis showed that primary site, pathological types, AJCC stage and surgery were independent factors affecting CSS. The C-index of the nomogram predicting CSS in training and validation set was 0.792 (average = 0.765, 95% CI [0.742–0.788]) and 0.776 (95% CI [0.773–0.779]), respectively. C-index based on nomogram was better in training and validation set than that based on AJCC stage. Calibration curves showed that these nomograms could accurately predict the 1-, 3- and 5-year OS and CSS both in training set and validation set.

**Conclusions.** The nomograms could effectively predict OS and CSS in young patients with PC, which help clinicians more accurately and quantitatively judge the prognosis of individual patients.

Corresponding author
Shu-Ping Yang, shupingy@163.com

# INTRODUCTION

PC is a highly malignant tumor. Global cancer data show that PC ranks 14th and 7th among all cancers in terms of incidence and mortality, respectively (*Bray et al., 2018*). It is estimated that by 2030, the mortality rate of PC will be the second highest in the world (*Rahib et al., 2014*). According to the American Cancer Society, in 2019, the United States will have 56,770 new cases of PC and 45,750 will die of PC. The 5-year survival rate of PC is the lowest of all cancers, with only 9% (*Siegel, Miller & Jemal, 2019*).

The incidence of PC in young patients is on the rise. A survey based on the SEER database found that the incidence of PC among young whites and blacks increased by 57% between 2001 and 2015 (*Tavakkoli et al., 2019*). In addition, another epidemiological study found that from 1992 to 2013, the annual percentage change in the incidence of PC in women aged 25–34 was more than 2.5% (*Gordon-Dseagu et al., 2018*). The median age of diagnosis of PC is 71 years old (*Midha, Chawla & Garg, 2016*). Due to ethnic and regional differences, the definition of " young patients " with PC has not been unified. Previous studies have generally classified young PC at the age of 45 (*Bunduc et al., 2018*; *Kang et al., 2017*; *He et al., 2013*) or 50 (*Ntala et al., 2018*; *Ansari et al., 2019*; *Beeghly-Fadiel et al., 2016*). In this study, we screened PC patients younger than 50 years old from the SEER database in order to enable more people to benefit from our study.

American Joint Committee on Cancer (AJCC) stage has always provided a reference for judging the progress of the disease and for the choice of clinical decision-making. However, the researchers found that the staging system is insufficient (*Adam et al., 2017*; *Choi et al., 2017*; *Shao et al., 2019*). It only incorporates some features of the tumors into the staging system, and the prognostic factors are far more than these. Previous studies have found that age, sex, grade, different treatment and even marital status are important independent factors affecting the prognosis of patients with PC (*Li, Zhou & Zhao, 2018*; *He et al., 2018*; *Wang et al., 2016*; *Baine et al., 2011*).

A nomogram is a graphical calculation or algorithm that combines several continuous variables to predict a specific end point using traditional statistical methods (for example, Logistic or Cox regression model) (*Kattan, 2002*). It has been constructed and verified in a variety of cancers to help clinicians quickly and accurately judge the prognosis of patients (*He et al., 2018*; *Kim et al., 2018*; *Zheng et al., 2018*). Many studies (*Bunduc et al., 2018*; *Kang et al., 2017*; *He et al., 2013*; *Ntala et al., 2018*; *Ansari et al., 2019*; *Beeghly-Fadiel et al., 2016*; *Eguchi et al., 2016*) have confirmed that the clinical characteristics and prognosis of young patients with PC are different from older patients, but the nomogram of PC in previous studies (*He et al., 2018*; *Song, Miao & Chen, 2018*; *Li & Liu, 2019*) cannot accurately predict the prognosis of young patients with PC.

The aim of this study was to establish and validate nomograms to predict 1-, 3-, and 5-year OS and CSS based on data from younger PC patients in the SEER database between 2004 and 2015.

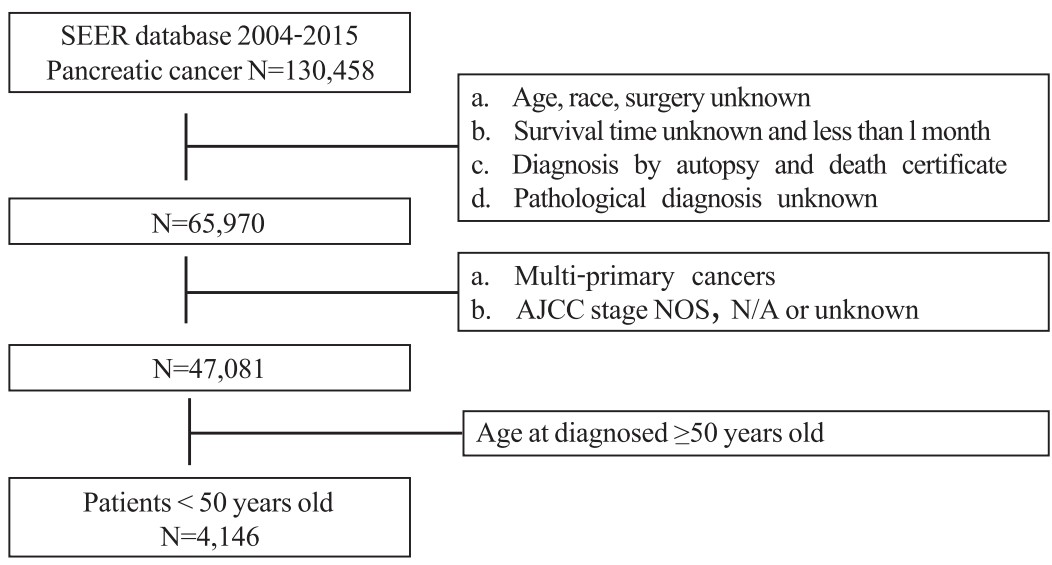

**Figure 1  Flowchart of patient selection.** Detailed selection of PC patients in 2004–2015 from SEER database.

# PATIENTS AND METHODS

## Selection of patients

PC patients younger than 50 years old who were diagnosed from January 1, 2004 to December 31, 2015 were selected from the Incidence-SEER 18 Registries Custom Data (with additional treatment fields), released April 2019, based on the November 2018 submission. The patient meted the following conditions: (1) site recode ICD-0-3/WHO 2008: ''pancreas''; (2) positive histology; (3) active follow-up; (4) one primary only. The exclusion criteria were as follows: (1) age, race, surgery unknown; (2) AJCC stage NOS, N/A or unknown; (3) age at diagnosed ≥50 years old; (4) survival time unknown and less than 1 month. The detailed flow chart was shown in Fig. 1. For this study, we signed the SEER research data agreement to access SEER information with the username10067-Nov2018. The SEER database is publicly available and the data for all patients are de-identified, so the approval and informed consent of the institutional review committee were not required in the current study.

## Variable classification

Clinical variables including race, gender, primary site, grade, pathological types, AJCC stage, status of surgery, survival time, status of survival and cause for death were extracted from the SEER database. The degree of differentiation of tumors was divided into three groups: grade I (well differentiated) and grade II (moderately differentiated) were high differentiated, grade III (poorly differentiated) and grade IV (undifferentiated) were low differentiated, and unknown. The pathological types were divided into ductal adenocarcinoma and non-ductal adenocarcinoma. The International Classification of Diseases for Oncology, Third Edition (ICD-0-3) codes of ductal adenocarcinoma were 8500 and 8140 (*Shi et al.,*

*2019*). The staging of cancer is based on the 6th edition of AJCC stage, which adapted to patients in the SEER database with a diagnosis time of 2004–2015.

## Statistical analysis

In this study, the whole cohort was randomly divided into two groups, 2,904 (70%) were training set and 1,242 (30%) were validation set. Chi-square test was used to compare the clinicopathological characteristics between the training set and the validation set.

OS referred to the duration from diagnosis to any original death or last follow-up. variables associated with OS in univariable analysis ($p < 0.05$) were included into multivariable analysis. The method of forward stepwise selection in a multivariable regression model was applied to the training cohort to select variables. The independent prognostic factors on multivariable analysis were used to construct nomogram for OS.

Considering that death from other causes is a competitive event of pancreatic cancer death, we constructed a competing risk nomogram to predict CCS. CSS referred to the duration from diagnosis to death from PC, patients who were alive at the point of last follow-up were considered as censored events. Variables related with CSS ($p < 0.05$) in the univariable analysis or with important clinical value were included into a multivariable analysis based on proportional subdistribution hazard models and those independent variables were selected to build a nomogram for CSS.

Discrimination of the nomograms was measured through the concordance index (C-index) with its respective 95% confidence interval (CI), which quantifies the level of concordance between probabilities of prediction and the actual chance of having the event of interest (*Iasonos et al., 2008*). The larger the C-index is, the more accurate the nomogram is to predict the prognosis (*Huitzil-Melendez et al., 2010*). 10-fold cross-validation was applied to verify the stability of the model and calculate the average value of C index in the training group. In order to reduce the overfit bias, calibration was evaluated by comparing the actual probabilities and the plot of the nomogram using 1,000 bootstrap samples. In the calibration curve, the vertical axis is the actual value and the horizontal axis is the predicted value. If the actual/predicted value passes through the origin along the 45° line, it shows that the nomogram has been well calibrated (*Iasonos et al., 2008*).

For all analyses, *p*-value <0.05 was considered statistically significant. All date was obtained through SEER*Stat software version 8.3.5. Fine and Gray competing risk Statistical analyses were performed using R work, and the others performed by SPSS (IBM, NY). The nomograms and calibration curves were draw using R version 3.6.0 (http://www.r-project.org)

## RESULTS

### Patients characteristic

Finally, there were a total of 4,146 cases of young patients with PC diagnosed in the SEER database between January 1, 2004 and December 31, 2015. In the entire cohort, in terms of demography, mainly whites (74.1%) and males (53.7%); In terms of tumor characteristics, pancreatic head cancer (46.1%) was the most common, more highly differentiated tumors (30.7%), mainly ductal adenocarcinoma (68.0%), in addition, more advanced tumors in

**Table 1  Clinical characteristics of training set and validation set.**

| Characteristics | | Total 4,146(100%) | Training set 2,904(70.0%) | Validation set 1,242(30.0%) | p-value |
|---|---|---|---|---|---|
| Race | | | | | 0.005 |
| | White | 3,073(74.1%) | 2,192(75.5%) | 881(70.9%) | |
| | Black | 666(16.1%) | 450(15.5%) | 216(17.4%) | |
| | Others | 407(9.8%) | 262(9.0%) | 145(11.7%) | |
| Gender | | | | | 0.761 |
| | Male | 2,225(53.7%) | 1,554(53.5%) | 671(54.0%) | |
| | Female | 1,921(46.3%) | 1,350(46.5%) | 571(46.0%) | |
| Primary site | | | | | 0.205 |
| | Head | 1,911(46.1%) | 1,355(46.7%) | 556(44.8%) | |
| | Body | 473(11.4%) | 327(11.3%) | 146(11.8%) | |
| | Tail | 791(19.1%) | 566(19.5%) | 225(18.1%) | |
| | Others | 971(23.4%) | 656(22.6%) | 315(25.4%) | |
| Grade | | | | | 0.393 |
| | High differentiated | 1,272(30.7%) | 909(31.3%) | 363(29.2%) | |
| | Low differentiated | 744(17.9%) | 520(17.9%) | 224(18.0%) | |
| | Unknown | 2,130(51.4%) | 1,475(50.8%) | 655(52.7%) | |
| Pathological types | | | | | 0.251 |
| | Ductal adenocarcinoma | 2,820(68.0%) | 1,991(68.6%) | 829(66.7%) | |
| | Non-ductal adenocarcinoma | 1,326(32.0%) | 913(31.4%) | 413(33.3%) | |
| AJCC | | | | | 0.833 |
| | I | 418(10.1%) | 297(10.2%) | 121(9.7%) | |
| | II | 1,098(26.5%) | 775(26.7%) | 323(26.0%) | |
| | III | 321(7.7%) | 228(7.9%) | 93(7.5%) | |
| | IV | 2,309(55.7%) | 1,604(55.2%) | 705(56.8%) | |
| Surgery | | | | | 0.256 |
| | No | 2,593(62.5%) | 1,800(62.0%) | 793(63.8%) | |
| | Yes | 1,553(37.5%) | 1,104(38.0%) | 449(36.2%) | |

diagnosis (55.7%); In terms of treatment, the proportion of patients undergoing surgery was low (37.5%). Detailed patient clinical characteristics were summarized in Table 1.

**Construction and validation of nomogram for predicting OS of young patients with PC**

The results of univariable and multivariable Cox regression models for OS were shown in Table 2. Univariable analysis showed that race, gender, primary site, grade, pathological types, AJCC stage and surgery were correlated with OS ($p < 0.05$). Multivariable analysis suggested that gender, grade, pathological types, AJCC stage and surgery were independent risk factors for OS. In details, female (HR:0.91, 95% CI [0.83–0.98]; $p = 0.019$), non-ductal adenocarcinoma (HR:0.40,95% CI [0.36–0.45]; $p < 0.001$) and receiving surgery (HR:0.43,95% CI [0.38–0.50]; $p < 0.001$) had a better OS. Black (HR:1.15, 95% CI [1.03–1.28]; $p = 0.016$), low differentiated tumor (HR:1.59, 95% CI [1.40–1.80]; $p < 0.001$) and advanced stage (HR:4.63, 95% CI [3.53–6.07]; $p < 0.001$) had a worse OS. The Nomogram

**Table 2 Univariate and multivariate COX analysis of OS in training set.**

| Characteristics | | Univariate analysis | Multivariate analysis | | |
|---|---|---|---|---|---|
| | | *p*-value | HR | 95% CI | *p*-value |
| Race | | 0.028 | | | 0.041 |
| | White | | Reference | | |
| | Black | | 1.15 | 1.03–1.28 | 0.016 |
| | Others | | 0.97 | 0.84–1.12 | 0.670 |
| Gender | | <0.001 | | | 0.019 |
| | Male | | Reference | | |
| | Female | | 0.91 | 0.83–0.98 | 0.019 |
| Primary site | | <0.001 | NA | | |
| | Head | | | | |
| | Body | | | | |
| | Tail | | | | |
| | Others | | | | |
| Grade | | <0.001 | | | <0.001 |
| | High differentiated | | Reference | | |
| | Low differentiated | | 1.59 | 1.40–1.80 | <0.001 |
| | Unknown | | 1.18 | 1.05–1.31 | 0.004 |
| Pathological types | | <0.001 | | | <0.001 |
| | Ductal adenocarcinoma | | Reference | | |
| | Non-ductal adenocarcinoma | | 0.40 | 0.36-0.45 | <0.001 |
| AJCC | | <0.001 | | | <0.001 |
| | I | | Reference | | |
| | II | | 3.05 | 2.23–3.96 | <0.001 |
| | III | | 3.10 | 2.30–4.18 | <0.001 |
| | IV | | 4.63 | 3.53–6.07 | <0.001 |
| Surgery | | <0.001 | | | <0.001 |
| | No | | Reference | | |
| | Yes | | 0.43 | 0.38-0.50 | <0.001 |

was constructed of the above six variables in training set (Fig. 2A). It can be seen from the nomogram that AJCC stage, which range of risk score is from 0 to 100, had the greatest contribution on OS (including 1-, 3- and 5-years), followed by pathological subtypes, surgery, grade, race and gender. The detailed steps for the application of the nomogram are as follows: Draw a vertical line to the horizontal axis marked "points" at the top of the nomogram according to the classification (e.g., gender is divided into male and female) of each prognostic variable (gender, grade, pathological type, AJCC stage, and surgery). At the position where the vertical line passes through the "Points" axis, each prognostic variable can obtain a score. Add the scores of the five variables to get the total score, find the position of the total score on the horizontal axis marked as "total points", and draw a vertical line from the total score position marked on the horizontal axis of "Total Points" to the 1-, 3- and 5-years OS axis. Where the vertical line intersects the 1-year OS axis is the % probability of the 1-year overall survival.

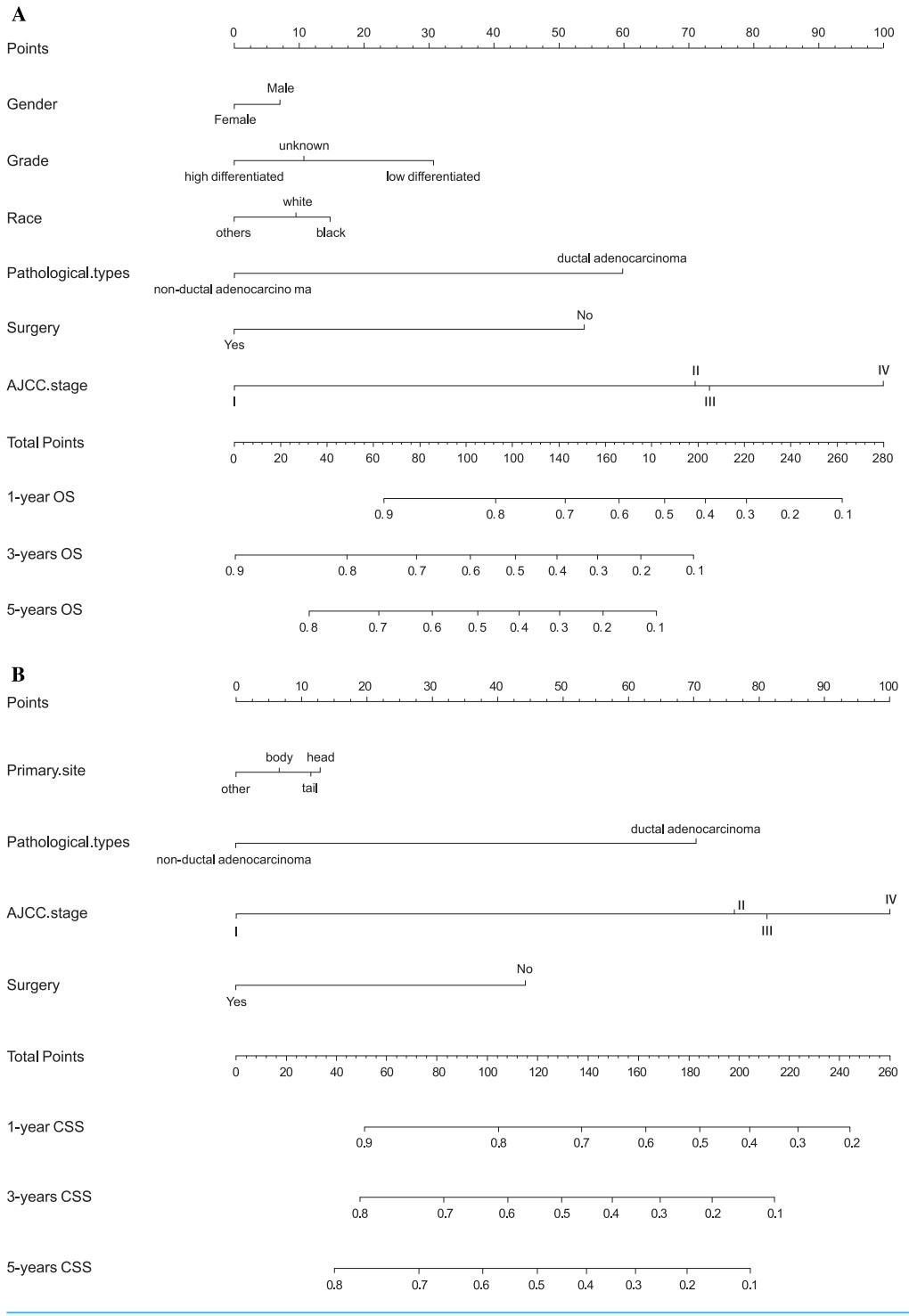

**Figure 2  Nomogram for predicting OS and CSS of young patients with PC.** (A) Nomogram for predicting 1-, 3- and 5-year OS for young patients with PC; (B) Nomogram for predicting 1-, 3- and 5-year CSS for young patients with PC.

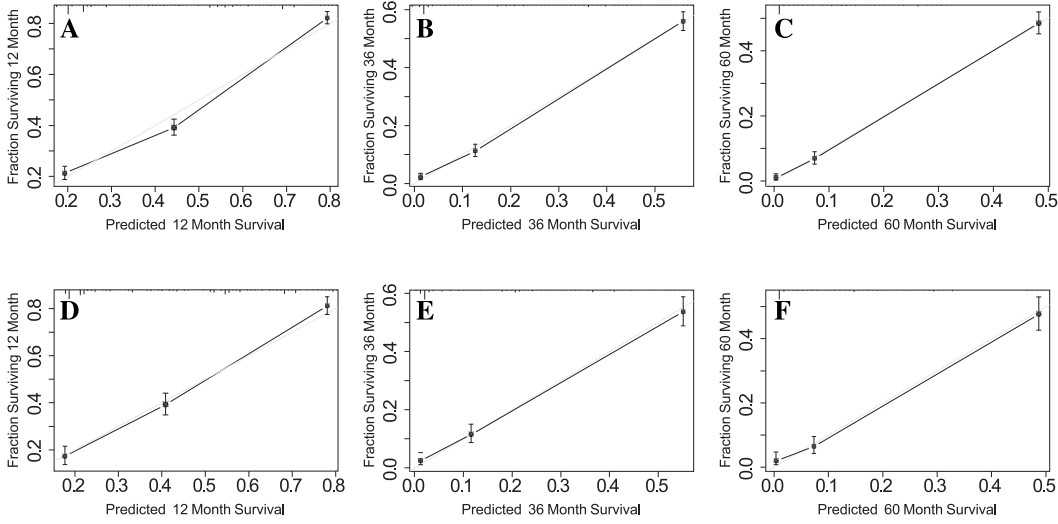

**Figure 3** **Calibration curves for 1-, 3- and 5-year OS in training set and validation set.** (A) Calibration curves for 1-year OS in training set; (B) Calibration curves for 3-year OS in training set; (C) Calibration curves for 5-year OS in training set; (D) Calibration curves for 1-year OS in validation set; (E). Calibration curves for 3-year OS in validation set; (F). Calibration curves for 5-year OS in validation set.

Compared with C-index based on AJCC stage in training set (0.677, 95% CI [0.666–0.688]) and validation set (0.672, 95% CI [0.656–0.688]), The C-index of our model in the training set and validation set was 0.733 (average = 0.731, 95% CI [0.724–0.738]) and 0.742 (95% CI [0.725–0.759]) respectively, showing a better degree of discrimination. The calibration curve of training set and validation set showed good consistency between prediction and observation in the probability of 1 -, 3 -, and 5-year OS, respectively (Fig. 3).

## Construction and validation of nomogram for predicting CSS of young patients with PC

The results of univariable and multivariable competing risks models for CSS were shown in Table 3. Fine and Gray analysis showed that primary site, grade, pathological types, AJCC stage and surgery were correlated with CSS. Multivariable analysis suggested that primary site, pathological types, AJCC stage and surgery were independent risk factors for CSS. In details, non-ductal adenocarcinoma (SHR:0.57, 95% CI [0.50–0.65]; $p < 0.001$), receiving surgery (SHR:0.77, 95% CI [0.68–0.87]; $p < 0.001$) and tumors in others site (SHR:0.87, 95% CI [0.77–0.99]; $p = 0.037$) had a better CSS. Advanced stage (SHR:1.53, 95% CI [1.22–1.93]; $p < 0.001$) had a worse CSS. The Nomogram was constructed of the above four variables in training set (Fig. 2.B). It also can be seen from the nomogram that AJCC stage had the greatest effect on CSS (including 1- ,3- and 5-years), followed by pathological subtypes, surgery and primary site. Finally, similar to the above, we could also predict 1 -, 3 -, and 5-year CSS in patients with PC.

Compared with C-index based on AJCC stage in training set (0.706, 95% CI [0.704–0.707]) and validation set (0.692, 95% CI [0.695–0.689]), The C-index of our model in the training set and validation set was 0.792 (average = 0.765, 95% CI [0.742–0.788]) and

**Table 3  Univariate and multivariate competing analysis of CSS in training set.**

| | | Univariate analysis | Multivariate analysis | | |
|---|---|---|---|---|---|
| | | p-value | SHR | 95% CI | p-value |
| Race | | | | | |
| | White | | Reference | | |
| | Black | 0.210 | 1.07 | 0.94–1.21 | 0.30 |
| | Others | 0.260 | 1.06 | 0.90–1.24 | 0.51 |
| Gender | | | | | |
| | Male | | Reference | | |
| | Female | 0.78 | 1.02 | 0.94–1.12 | 0.63 |
| Primary site | | | | | |
| | Head | | Reference | | |
| | Body | 0.046 | 1.01 | 0.88–1.17 | 0.86 |
| | Tail | 0.091 | 1.09 | 0.96–1.24 | 0.18 |
| | Others | 0.750 | 0.87 | 0.77–0.99 | 0.037 |
| Grade | | | | | |
| | High differentiated | | Reference | | |
| | Low differentiated | 0.024 | 1.09 | 0.96–1.23 | 0.17 |
| | Unknown | <0.001 | 1.09 | 0.97–1.21 | 0.14 |
| Pathological types | | | | | |
| | Ductal adenocarcinoma | | Reference | | |
| | Non-ductal adenocarcinoma | <0.001 | 0.57 | 0.50–0.65 | <0.001 |
| AJCC | | | | | |
| | I | | Reference | | |
| | II | 0.063 | 1.22 | 0.99–1.51 | 0.072 |
| | III | <0.001 | 1.26 | 0.98–1.61 | 0.069 |
| | IV | <0.001 | 1.53 | 1.22–1.93 | <0.001 |
| Surgery | | | | | |
| | No | | Reference | | |
| | Yes | <0.001 | 0.77 | 0.68–0.87 | <0.001 |

0.776 (95% CI [0.773–0.779]), respectively, showing a better degree of discrimination. The calibration curve of training set and validation set also showed good consistency between prediction and observation in the probability of 1 -, 3 -, and 5-year CSS, respectively (Fig. 4).

## DISCUSSION

In this study, we determined that race, gender, grade, pathological types, surgery and AJCC stage were independent factors affecting the OS of younger patients with PC by univariable and multivariable regression analysis based on the SEER database, and competing risk analysis was used to determine that surgery, pathological types and AJCC stage and primary site were independent factors affecting CSS in younger patients with PC. We integrated the factors and draw nomograms that could effectively predict 1 -, 3 -, and 5-year OS and CSS in younger patients with PC.

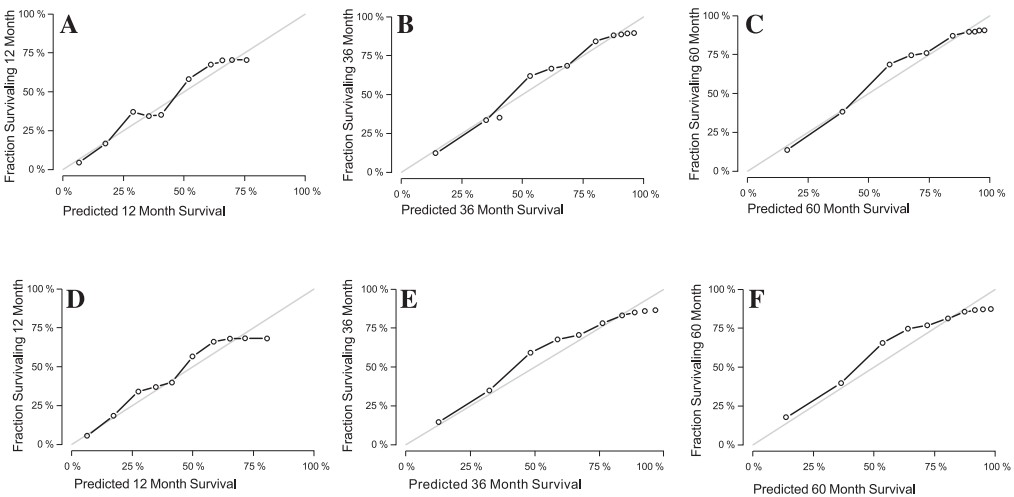

**Figure 4** **Calibration curves for 1-, 3- and 5-year CSS in training set and validation set.** (A) Calibration curves for 1-year CSS in training set; (B). Calibration curves for 3-year CSS in training set; (C). Calibration curves for 5-year CSS in training set; (D). Calibration curves for 1-year CSS in validation set; (E). Calibration curves for 3-year CSS in validation set; (F). Calibration curves for 5-year CSS in validation set.

As far as we know, this was the first time that a nomogram based on a large multicenter dataset has been constructed to effectively predict the prognosis of younger patients with PC. Previous studies (*He et al., 2018*; *Song, Miao & Chen, 2018*; *Li & Liu, 2019*; *Pu et al., 2018*) found that age was an independent factor affecting the OS or CSS of patients with PC, and most of them divided age into <60 years (or 65 years) and $\geq$ 60 years (or 65 years). There were differences in clinicopathological characteristics and prognosis between young patients with PC and elderly patients (*Bunduc et al., 2018*; *Kang et al., 2017*; *He et al., 2013*; *Ntala et al., 2018*; *Ansari et al., 2019*; *Beeghly-Fadiel et al., 2016*; *Eguchi et al., 2016*). Our data showed that primary site was an independent factor affecting CSS in young patients with PC, which was consistent with previous study (*Song, Miao & Chen, 2018*). Another study[29] used five variables, including age, differentiation, TNM staging, surgery and lymph node surgery, to construct a nomogram for predicting the OS rate of patients with PC, of which age was divided into 25–39, 40–59, 60–79 and 80 +. But the study still had some limitations. They included only patients with non-metastasis and ductal adenocarcinoma. Our study found that in young patients with PC, patients with distant metastasis and non-ductal adenocarcinoma accounted for about 55.7% and 32.0% of the total population, respectively. Therefore, it was necessary to establish a nomogram that could predict the prognosis of young patients with PC.

In our study, AJCC stage and surgery were independent factors affecting OS and CSS in young patients with PC, which was consistent with the results of previous studies (*He et al., 2018*; *Song, Miao & Chen, 2018*; *Pu et al., 2018*) in patients with PC. AJCC stage had the greatest influence on OS and CSS in young patients with PC, followed by pathological type, surgery and grade. Additionally, it was worth noting that the impact of pathological types on prognosis was even greater than that of surgery. *Mostafa et al. (2017)* pointed out that

although ductal adenocarcinoma was the most common pathological type of PC, there were many other pathological types of PC in the real world, such as solid pseudopapillary tumors, neuroendocrine tumors, and so on. They had different clinicopathological characteristics and biological behavior. Therefore, more research was needed to focus on non-ductal adenocarcinoma. There were racial differences in the OS of young patients with PC, and the risk of death was higher in blacks. This might be due to the higher incidence of distant pancreatic cancer in black patients (*Tavakkoli et al., 2019*).

Compared with the traditional AJCC stage, all the C-index of our model were more than 0.72, indicating a better discrimination and the ability to provide individualized prediction for patients. For example, two patients with stage II PC after surgery, one was a male with low differentiated ductal adenocarcinoma, and the other was a female with high differentiated non-ductal adenocarcinoma. According the Table S1, the 1-year OS of the two patients could be calculated to be about 56% and 88%, respectively. However, according to the AJCC stage, both of them were patients with stage II PC, and it was difficult to compare the differences in prognosis between them.

Our research had some advantages. First of all, our study was based on the SEER database, which covered 28% of the population of the United States, so the nomograms were more applicable. Secondly, compared with the previous nomograms used to evaluate the prognosis of PC patients, our models were more targeted to evaluate the prognosis of PC patients younger than 50 years old. Finally, the calibration curves of our training set and validation set showed good consistency, indicating that our nomograms had good calibration power.

Even so, some limitations still existed in our study. First, at present, surgery was still the main treatment for patients with PC, but studies have found that radiotherapy and chemotherapy can also effectively improve the prognosis of postoperative patients with PC (*Springfeld et al., 2019*; *Stessin, Meyer & Sherr, 2008*). Detailed radiotherapy or chemotherapy regimens were not available from the SEER database. In addition, other known risk factors for prognosis, such as family history (*Schulte et al., 2016*), tobacco or alcohol *Korc et al. (2017)* were also hard to obtain from the SEER database. Moreover, in this study, patients were randomly divided into two groups, 70% of them were used to construct and other 30% were used to validate the nomograms. Although both the C index and the calibration curve performed well, external validation was still needed in other populations to evaluate the accuracy of our models.

## CONCLUSION

We developed and validated nomograms that could effectively predict the prognosis of PC patients younger than 50 years old. These nomograms could help clinicians more accurately and conveniently predict the 1-, 3-and 5-year OS and CSS of individual patients.

**Abbreviations**

| | |
|---|---|
| **PC** | pancreatic cancer |
| **OS** | overall survival |
| **CSS** | cancer-specific survival |

| **SEER** | Surveillance, Epidemiology, and End Results |
| **C-index** | concordance index |
| **AJCC** | American Joint Committee on Cancer |

## ACKNOWLEDGEMENTS

The authors acknowledge the efforts of the Surveillance, Epidemiology, and End Results Program tumor registries in the creation of the SEER database.

### Funding

The authors received no funding for this work.

### Competing Interests

The authors declare there are no competing interests.

### Author Contributions

- Min Shi performed the experiments, analyzed the data, prepared figures and/or tables, authored or reviewed drafts of the paper, and approved the final draft.
- Biao Zhou analyzed the data, prepared figures and/or tables, and approved the final draft.
- Shu-Ping Yang conceived and designed the experiments, authored or reviewed drafts of the paper, and approved the final draft.

### Data Availability

The raw data and the R code for analysis and the step of click in SPSS are available as Supplementary Files.

### Supplemental Information

Supplemental information for this article can be found online at http://dx.doi.org/10.7717/peerj.8958#supplemental-information.

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
