# Peer review of "Nomograms for predicting overall survival and cancer-specific survival in young patients with pancreatic cancer in the US based on the SEER database"

_PeerJ, doi:10.7717/peerj.8958_

## Round 0.1 · original submission · Major Revisions

I applaud the authors' effort to provide as much details as possible. However, I believe the manuscript can be further enriched by addressing the comments from the two reviewers.

Reviewer 1 ·

Basic reporting

I suggest to change the tense of the first sentence of the abstract from "The incidence of young patients with pancreatic cancer (PC) was on the rise, and there was a lack of models that could effectively predict their prognosis." to "The incidence of young patients with pancreatic cancer (PC) is on the rise, and there is a lack of models that could effectively predict their prognosis."

Line 54-55, suggest changing "inadequate" to "insufficient".

Citation format needs reviewing throughout the manuscript e.g. lines 54-55, 59, 62.

A more detailed explanation of what a nomogram is would benefit readers.

Please change "COX risk regression model" to "Cox regression model" or "Cox proportional hazards model", e.g. line 97-98.

A citation on how the C-index is defined or how it is calculated would also be beneficial for an average reader.

Experimental design

The selection criteria and exclusions are clearly stated.

There seems to be a typo of the diagnosis time in line 93, please clarify.

Validity of the findings

Can the authors comment whether they consider patients who died from a different cause other than PC as censored? In other words, is there a reason authors didn't use a more standard competing risk strategy for their CSS analysis?

Regarding multivariate analysis, further explanation on how the authors decided which variables to include would be appreciated. Specifically, did the authors use any kind of step-wise/p-value approach?

Additional comments

The authors conducted a thorough multi-center retrospective study to identify prognostic factors for OS and CSS in young patients with PC. Moreover, the authors utilize nomograms to ease interpretation in the clinical setting and beyond. I applaud the authors' effort to provide as much details as possible. However, I believe the manuscript can be further enriched by addressing the comments/suggestions I'm raising.

Reviewer 2 ·

Basic reporting

1. In Figures 3 and 4, the x-axis labels, y-axis labels, x-tick labels and y-tick labels are not clear. It is strongly suggested that the authors adjust the labels appropriately.

2. In Line 149 of the Results Section, it is not intelligible in the description that “it also can be seen from the nomogram that AJCC stage had the greatest effect on prognosis.” Which OS does the “prognosis” denote, 1-year, 3-year, or 5-year OS? Please explain more details in how to interpret the nomograms and how to evaluate the effect of the variables on the specific prognosis outcome from the nomograms directly.

3. The authors should provide the raw data and the scripts (or instructions of point and click in SPSS) used to perform all analyses, including how to generate the nomograms and calibration curves.

Experimental design

No comment

Validity of the findings

1. In Line 95 of the Statistical Analysis Section, the authors mentioned that the whole cohort was randomly divided into the training and validation groups. Do the authors try another random split of the whole cohort and evaluate the results? How about the results of 10-fold cross-validation? Are the current results consistent to the results of cross-validation?

Additional comments

No comment.

---

## Round 0.2 · Minor Revisions

Dear Dr. Shi,

Thank you for your submission to PeerJ. The re-reviewers have commented on your revision and as you can see Reviewer 1 in particular has some serious concerns and has suggested that you use a more complex model. Please address these concerns in full.

Reviewer 1 ·

Basic reporting

Change "univariate" for "univariable" throughout the manuscript
Change "mutivariate" for "multivariable" throughout the manuscript
Page 8, line 148: change "COX" to "Cox"
Page 9, line 179: change "competing models" to "competing risks models"
Page 11, line 215: change "precious" to "previous"

Experimental design

no comment

Validity of the findings

Using only significant results in univariable models into the multivariable models is an inadequate strategy for variable selection and is not recommended. The main reason is that this selection strategy fails to include potentially meaningful predictors or even well-known confounders if deemed non-significant. Thus, a more robust or clinically-guided model is often preferred. Have the authors analyzed a more comprehensive multivariable model with most (if not all) available factors in the SEER database to evaluate the prognostic value of this "comprehensive" model? I understand the rationale of constructing a more parsimonious model, however, the approach you used for variable selection may have left out potentially meaningful predictors such as gender in the CSS analysis. I believe that given the sample size in the database, fitting a slightly more complex model won't represent a major issue.

Additional comments

I appreciate the work the authors have put to improve their work during the first round of revision. Significant changes and refinements were made as requested. Considering the explanations on how the multivariable models were constructed major comments.

Reviewer 2 ·

Basic reporting

The description of the nomogram method is clear and straightforward.

The displaying of figures and tables is proper and clear to readers.

Experimental design

The selection criteria and exclusions are clearly stated.

Validity of the findings

Can the author describe the results of 10-fold cross-validation and the calibration between the actual probabilities and the plot of the nomogram in Line 116-118?

Additional comments

The authors suggest nomograms which can help interpret the results more easily in clinical setting. I appraise the authors' effort to explore the independent factors associated to PC using different analysis methods and to provide as much details as possible.

---

## Round 0.3 · accepted · Accept

Dear Dr. Shi,

Thank you for your submission to PeerJ.

I am writing to inform you that your manuscript - Nomograms for predicting overall survival and cancer-specific survival in young patients with pancreatic cancer in the US based on the SEER database - has been Accepted for publication. Congratulations!

Your article is being checked and you will receive a list of Production tasks shortly.

Once your production tasks are completed, your proofing PDF will be created (please do not proof check your reviewing PDF!).